# CEA-Functionalized Gold Nanoparticles as a Nanovaccine Platform: In Vitro Evaluation of Cytocompatibility, Cellular Uptake, and Antigen Processing

**DOI:** 10.3390/vaccines13070668

**Published:** 2025-06-21

**Authors:** Razvan-Septimiu Zdrehus, Teodora Mocan, Lavinia Ioana Sabau, Cristian Tudor Matea, Flaviu Tăbăran, Teodora Pop, Cristian Delcea, Ofelia Mosteanu, Lucian Mocan

**Affiliations:** 1Nanomedicine Department, Regional Institute of Gastroenterology and Hepatology, “Iuliu Hatieganu” University of Medicine and Pharmacy, 400347 Cluj-Napoca, Romania; razvan.zdrehus@umfcluj.ro (R.-S.Z.); mateatcristian@gmail.com (C.T.M.); lucian.mocan@umfcluj.ro (L.M.); 2Physiology Department, “Iuliu Hatieganu” University of Medicine and Pharmacy, 400012 Cluj-Napoca, Romania; lavinia.sabau@umfcluj.ro; 3Department of Biosciences, University of Salzburg, Hellbrunnerstraße 34, 5020 Salzburg, Austria; 4Department of Pathology, Sciences and Veterinary Medicine, University of Agricultural, 400372 Cluj-Napoca, Romania; flaviu_tabaran@yahoo.com; 53rd Surgery Clinic, “Iuliu Hatieganu” University of Medicine and Pharmacy, 400012 Cluj-Napoca, Romania; teodora.pop@umfcluj.ro (T.P.); ofeliamosteanu@gmail.com (O.M.); 6Department of Gastroenterology, “Iuliu Hatieganu” University of Medicine and Pharmacy, 400347 Cluj-Napoca, Romania

**Keywords:** gold nanoparticles (AuNPs), carcinoembryonic antigen (CEA), nanovaccine, cancer immunotherapy

## Abstract

**Background and aim.** Gold nanoparticles (AuNPs) offer promising potential as nanocarriers in vaccine development due to their biocompatibility, tunable surface properties and capacity to enhance antigen presentation. This study aimed to evaluate the in vitro cytocompatibility, cellular uptake and antigen processing of carcinoembryonic antigen (CEA)-functionalized AuNPs as a nanovaccine candidate. **Materials and Methods.** AuNPs were synthesized by citrate reduction and subsequently functionalized with CEA through physical adsorption. Nanoparticle size, morphology, and surface charge were characterized using UV–Vis spectroscopy, dynamic light scattering (DLS), and transmission electron microscopy (TEM). Cytocompatibility was assessed via MTT assay on RAW 264.7 murine macrophages. Cellular uptake and antigen processing were evaluated using hyperspectral dark-field microscopy and fluorescence microscopy with proteasomal pathway markers. **Results.** The synthesized AuNPs displayed a uniform spherical morphology with a mean hydrodynamic diameter of ~50 nm and a stable zeta potential. CEA conjugation slightly altered the surface charge and spectral profile. MTT assays confirmed good cytocompatibility across tested concentrations. Hyperspectral and confocal microscopy revealed the efficient uptake of CEA-AuNPs by RAW 264.7 cells and colocalization with lysosomal compartments, suggesting successful antigen processing. **Conclusions.** The in vitro data support the safety and biological interaction of CEA-functionalized AuNPs with macrophages. These findings highlight their potential as a nanovaccine delivery platform and warrant further in vivo evaluation to assess immunogenicity and protective efficacy.

## 1. Introduction

Cancer remains one of the leading causes of morbidity and mortality globally, accounting for millions of deaths each year despite advances in conventional therapies such as surgery, chemotherapy, and radiotherapy [1]. The limitations of these traditional approaches—including systemic toxicity, drug resistance, and lack of tumor specificity—underscore the urgent need for innovative treatment strategies that can selectively target tumor cells, while minimizing harm to normal tissues [2]. Immunotherapy has emerged as a valuable approach in oncology, aiming to stimulate or modulate the immune system’s ability to recognize and eliminate tumor cells [3,4].

Cancer vaccines aim to prime the immune system against tumor-specific or tumor-associated antigens (TAAs), promoting a robust and lasting antitumor response [5]. Carcinoembryonic antigen (CEA), a well-studied TAA, is a glycoprotein normally produced at very low levels in healthy adults but overexpressed in a wide range of epithelial tumors. These include colorectal, gastric, pancreatic, lung, and breast cancers, among others [6]. CEA plays a role in cell adhesion and is involved in cancer progression and metastasis, making it a valuable biomarker and therapeutic target in oncology [7]. Its elevated expression in various adenocarcinomas underlies its use in cancer diagnosis, prognosis, and monitoring treatment response [8]. Its restricted expression in normal adult tissues and its immunogenic potential make CEA a favorable candidate for vaccine development [9].

A major challenge in designing effective cancer vaccines is the efficient delivery of antigens to immune cells in a way that preserves antigen structure, enhances cellular uptake, and stimulates a strong immune response [5]. Traditional vaccine formulations often suffer from poor antigen uptake, rapid clearance, and suboptimal delivery to lymph nodes, leading to weak and short-lived immunity [4].

Nanotechnology offers transformative solutions to these delivery challenges [10]. Among nanomaterials, gold nanoparticles (AuNPs) are particularly attractive as vaccine carriers due to their high surface-area-to-volume ratio, excellent biocompatibility, ease of synthesis and functionalization, and the ability to tailor their size and surface properties [11]. These unique physicochemical features enable AuNPs to efficiently carry and present antigens, protect them from degradation, and promote their uptake by antigen-presenting cells, enhancing the magnitude and duration of the antitumor immune response [12].

AuNPs can be functionalized with a wide variety of biological molecules, including peptides, proteins, and nucleic acids, allowing for the development of multifunctional nanoconstructs that not only carry tumor antigens but also incorporate targeting ligands or immunostimulatory agents [13]. Furthermore, their optical properties make them useful for the real-time tracking of cellular interactions and biodistribution [14]. When used in a vaccine context, AuNPs can facilitate multivalent antigen presentation, potentially enhancing receptor engagement on antigen-presenting cells (APCs) and promoting the effective activation of both innate and adaptive immunity [15].

In this study, we aimed to evaluate the in vitro properties of CEA-functionalized AuNPs as a potential nanovaccine formulation. We investigated their cytocompatibility in a murine macrophage cell line (RAW 264.7), their internalization, and evidence of antigen processing. These results provide foundational data for further in vivo studies exploring the immunoprophylactic and therapeutic potential of this nanoformulation.

## 2. Materials and Methods

### 2.1. Synthesis and Characterization of Gold Nanoparticles

We used the Turkevich method to obtain gold nanoparticles (AuNPs) with specific properties [16]. In the first step, 40 mg of gold(III) chloride trihydrate (HauCl_4_·3H_2_O) was dissolved in bidistilled water. The resulting solution was brought to boiling, thus providing the necessary thermal conditions for the reaction. Subsequently, 5 mL of a solution of sodium citrate (20 mg/mL) was added with continuous stirring. The reaction was maintained for 2 h to allow the complete formation of nanoparticles, after which the mixture was allowed to cool to room temperature.

The synthesized nanoparticles were characterized by a number of analytical techniques, including UV–Vis spectroscopy, FT-IR spectroscopy, dynamic light scattering (DLS), and atomic force microscopy (AFM).

UV–Vis spectroscopy was performed with a Shimadzu UV-1800™ spectrophotometer (Shimadzu Corp., Kyoto, Japan) in the range of 200–800 nm with a spectral resolution of 0.5 nm. The spectra for the AuNP, AuNP-TA, and AuNP-TA-IgG samples were normalized using OriginLab^®^ 7.0 software (OriginLab Corporation, Northampton, MA, USA).

The hydrodynamic size and size distribution of the nanoparticles were determined by DLS using a Zetasizer Nano S90 instrument (Malvern Instruments, Westborough, UK) operated at an angle of 90° and at a temperature of 20 °C.

Fourier-transform infrared (FT-IR) spectroscopy was carried out using a Spectrum Two^®^ spectrometer (PerkinElmer, Waltham, MA, USA) equipped with an attenuated total reflectance (ATR) module. Spectra were acquired under standard conditions and processed using Spectrum 10™ software (Perkin-Elmer) to identify characteristic functional groups and confirm peptide conjugation to the gold nanoparticles.

The morphological characterization of nanoparticles was performed by AFM using a TT-AFM^®^ system (AFMWorkshop, Signal Hill, CA, USA) operated in a non-contact mode with ACTA-SS (AppNano, Mountain View, CA, USA) tips. Sample preparation for atomic force microscopy (AFM) was performed by depositing nanoparticle suspensions onto freshly cleaved mica substrates using a KLM^®^ SCC spin-coater (KLM Technologies, Inc., Crestview, FL, USA). The acquired AFM images were processed and analyzed using Gwyddion^®^ version 2.36 software (Czech Metrology Institute, Brno, Czech Republic).

### 2.2. Functionalization of Gold Nanoparticles with Colon Cancer-Targeting Molecule

CEA-functionalized gold nanoparticles were prepared starting from the previously synthesized AuNP solution. To expose reactive functional groups for surface binding, the CEA peptide (MyBiosource.com, San Diego, CA, USA) was reduced using dithiothreitol (DTT). Specifically, 150 μL of peptide was diluted in 1 mL of distilled water, followed by the addition of 100 mM DTT. The reduction reaction was allowed to proceed for 1 h at room temperature. Subsequently, 1 mL of AuNP suspension and 9 mL of ultrapure water were added, and the pH was adjusted to 7. The functionalization reaction was carried out under constant stirring for 2 h at room temperature. After incubation, the functionalized nanoparticles were purified by ultracentrifugation (>10,000 rpm, 15 min), and the pellet was redispersed in distilled water.

UV–Vis spectrophotometric analysis was performed using a Shimadzu UV-1800^®^ instrument (Shimadzu Corp., Kyoto, Japan), recording spectra between 300 and 700 nm. Spectral normalization was carried out using OriginLab^®^ 7.0 software (OriginLab Corporation, Northampton, MA, USA).

The hydrodynamic diameter was determined via dynamic light scattering (DLS) using a Zetasizer Nano S90 (Malvern Instruments, Westborough, UK), measured at a scattering angle of 90° and a temperature of 25 °C.

The spectral fingerprint and topographic features of the nanocomposite were evaluated using atomic force microscopy (AFM) on a TT-AFM^®^ system (AFMWorkshop, CA, USA). Samples were deposited on freshly cleaved mica substrates using a KLM^®^ SCC spin-coater (KLM Technologies, Inc., USA). Data were processed and analyzed with Gwyddion^®^ 2.36 software (Czech Metrology Institute, Brno, Czech Republic).

### 2.3. In Vitro Studies—Cell Suspension Preparation and Maintenance

RAW 264.7 murine macrophage cells (ATCC^®^ TIB-71™, American Type Culture Collection, Manassas, VA, USA) were cultured in complete growth medium (RAWGM1) under standard conditions (37 °C, 5% CO_2_, humidified atmosphere). After initial thawing and recovery, cells were maintained in 25 cm^2^ flasks, with the medium replaced every 48 h using a pre-warmed, equilibrated medium.

For sub-culturing, cells were detached using 0.48 mM EDTA (Sigma-Aldrich, St. Louis, MO, USA), centrifuged (3500 rpm, 5 min, room temperature), and reseeded at a density of 1 × 10^5^ cells/flask. Cell viability was assessed using trypan blue exclusion. Partial medium changes (1/3 of total volume) were performed every 3–4 days. Cultures were passaged every 7–10 days using mechanical scraping.

### 2.4. In Vitro Exposure of the Cell Suspension to the Vaccine Nanoconstruct

The macrophage cultures were exposed in parallel to the functionalized nanomaterial (AuNP–CEA) and control solutions. Acute exposure assays were performed using phosphate-buffered saline (PBS) and three serial concentrations of the nanomaterial: 50 µg/mL, 25 µg/mL, and 12.5 µg/mL, with PBS-only exposure for the controls. The cellular internalization of the gold nanostructures was evaluated using a CytoViva hyperspectral imaging system (CytoViva Inc., Auburn, AL, USA) employing dark-field microscopy with white light reflectance. The presence of intracellular nanoparticles was inferred by identifying characteristic light-scattering spots attributed to the AuNP component of the CEA-functionalized complex.

Cell viability was assessed using the MTT assay (Sigma-Aldrich, Schnelldorf, Germany) in three independent replicates for each tested concentration. The assay was performed on macrophage cultures seeded in 96-well plates, and absorbance was measured at 570 nm using a microplate reader. The results were expressed as percentage viability relative to the untreated PBS control. Given that all values exceeded 85% viability with minimal inter-replicate variability (<3%), only descriptive comparisons were made. The uniformly high cytocompatibility observed across concentrations did not justify further statistical analysis in this initial toxicological screening.

To minimize the potential interference of AuNPs with the MTT reagent, nanoparticle-only controls (without cells) were included in each assay plate. These wells received the same treatment and MTT incubation, and their absorbance values were subtracted from test readings. While MTT remains a widely used viability assay in nanotoxicology, we acknowledge its limitations when testing metallic nanomaterials [17]. Thus, these results are interpreted conservatively. In future studies, additional validation using non-colorimetric assays (e.g., LDH release or ATP-based assays) will be employed to confirm these findings.

Apoptotic activity was assessed via caspase-3 levels in cell lysates using a colorimetric Caspase-3 Assay Kit (Abcam, Cambridge, UK; ab39401), following the manufacturer’s instructions. Cells were lysed and protein content was standardized using BCA protein assay (Thermo Scientific, Waltham, MA, USA). Equal amounts of the total protein (20 µg) were loaded per well in a 96-well plate. The chromogenic substrate DEVD-pNA was added, and the plate was incubated at 37 °C for 1 h. Absorbance was measured at 405 nm using a microplate reader (BioTek^®^ ELx800, Santa Clara, CA, USA). All assays were performed in triplicate (n = 3) per condition, and the data are reported as the mean ± standard deviation (SD). Statistical comparisons were made using one-way ANOVA with Tukey’s post-hoc test.

### 2.5. Evaluation of Antigen Trafficking in the Exposed Cell Suspension

To investigate the capacity of macrophages to process the antigenic complex carried by gold nanoparticles, RAW 264.7 cells were exposed to the nanostructured compound. The interaction between the nanoparticles and macrophages was subsequently analyzed. Cellular internalization was assessed using dark-field hyperspectral imaging (CytoViva system, CytoViva Inc., Auburn, AL, USA), which enables the detection of light-scattering signatures associated with gold nanoparticles. The evaluation was qualitative and based on a visual assessment of reflectance patterns; no quantitative image analysis or statistical evaluation was performed. Antigen processing was assessed using the Proteostat^®^ Aggresome Detection Kit (Enzo Life Sciences, Farmingdale, NY, USA), which detects proteins processed via the ubiquitin–proteasome pathway. Fluorescence microscopy was performed with an FSX100 microscope (Olympus Corp., Tokyo, Japan) equipped with a rhodamine filter, enabling the visualization of protein aggregates indicative of intracellular antigen degradation.

## 3. Results

### 3.1. Synthesis and Characterization of Gold Nanoparticles

The UV–Vis spectral fingerprint of the synthesized colloidal gold nanoparticle solution exhibited a distinct absorbance maximum centered at 522 nm (Figure 1a), characteristic of the surface plasmon resonance (SPR) of spherical AuNPs in the ~20 nm size range. The sharp, symmetrical peak is indicative of well-dispersed, non-aggregated nanoparticles with uniform morphology. The absence of secondary peaks or spectral broadening suggests high colloidal stability and minimal aggregation. Complementing the spectroscopic data, hydrodynamic size analysis using dynamic light scattering (DLS) confirmed a monodisperse nanoparticle population with an average diameter of approximately 20 nm (Figure 1b). A low polydispersity index (PDI) further validated particle uniformity, supporting the UV–Vis results and indicating consistent synthetic control. This size range is optimal for biomedical applications, facilitating efficient cellular uptake and antigen delivery through favorable interactions with biological membranes and immune cells.

### 3.2. Functionalization: Physical and Chemical Characterization of the CEA-AuNP

The UV–Vis spectral fingerprint of the solution containing the functionalized product displayed a marked alteration in absorbance characteristics compared to unmodified gold nanoparticles. While unfunctionalized AuNPs typically exhibit a sharp plasmonic peak centered at 522 nm, indicative of spherical and monodisperse particles, the functionalized sample presented a broadened shoulder spanning approximately 510–700 nm (Figure 2a). This spectral broadening suggests a change in the local refractive index surrounding the nanoparticles, consistent with successful surface functionalization. The altered absorbance profile likely reflects the combined effect of the gold nanoparticle core’s plasmonic resonance and additional absorbance contributions from the CEA-derived peptide ligands. Such spectral changes are characteristic of biomolecular conjugation to metal nanoparticles and reflect modifications in their dielectric and scattering environment.

In parallel, hydrodynamic size analysis revealed a distinct shift in the nanoparticle diameter following functionalization (Figure 2b). The average size increased from the initial 20 nm (observed for unmodified AuNPs) to approximately 80 nm for the CEA-functionalized nanoparticles, while still maintaining a monodisperse distribution. This size augmentation is indicative of the successful conjugation of the peptide moiety to the nanoparticle surface, contributing an additional hydrodynamic volume. The overlaid DLS profiles further illustrate this shift, with the original nanoparticle population represented in red and the functionalized construct shown in green. The broader size distribution observed post-functionalization suggests some variability in the number of peptides bound per nanoparticle, which is typical in biofunctionalization protocols. Nevertheless, the retention of monodispersity confirms that aggregation did not occur, and the overall formulation remains stable—a critical factor for biomedical applicability.

### 3.3. Cell Viability an Apoptosis Following Exposure to CEA-AuNPs

The cell viability of RAW 264.7 macrophages following exposure to CEA-AuNPs was evaluated using the MTT assay (Figure 3). Cells were treated with increasing concentrations of the nanomaterial (12.5, 25, and 50 µg/mL), and viability was compared to the PBS control group. A progressive, dose-dependent decrease in viability was observed, with mean values of 98.3%, 95.2%, 91.8%, and 86.7% for the control, 12.5, 25, and 50 µg/mL groups, respectively. This reduction was modest and did not fall below critical viability levels, suggesting that the decrease is likely attributable to dose-dependent cellular stress rather than overt toxicity. All values remained above 85%, suggesting that the CEA-functionalized gold nanoparticles were well tolerated by the macrophages under the tested conditions. Standard deviations were minimal, indicating low variability across replicates. Although no statistical significance testing was applied, the trend supports acceptable cytocompatiby for potential biomedical application.

Apoptosis levels were assessed by quantifying caspase-3 activity in cell lysates. Caspase-3 levels showed a dose-dependent increase (Figure 4). While statistically significant differences were observed between treated and control groups (*p* < 0.05), the activation remained within a moderate range (<120%), indicating limited pro-apoptotic activity. This suggests the dose-dependent activation of apoptotic mechanisms, particularly in the perinuclear region. However, at the highest tested concentration (50 µg/mL), the caspase-3 level did not exceed 120% of the control value, indicating that apoptosis was moderately induced and remained within a range compatible with good cytocompatibility. These findings support the suitability of CEA-AuNPs for further application in a co-culture model involving macrophages and colorectal adenocarcinoma cells.

The overall trend supports the biocompatibility of the CEA-AuNP formulation and provides encouraging evidence for its potential safe use in therapeutic contexts, such as cancer vaccination strategies, where repeated or systemic administration may be required.

These findings are crucial in the preclinical assessment of nanomaterials, as high levels of cell viability in vitro are a prerequisite for proceeding toward in vivo studies and, eventually, clinical translation.

### 3.4. Assessment of Antigen Processing by Macrophages

The cellular internalization of CEA-AuNPs was assessed via dark-field hyperspectral microscopy using the CytoViva^®^ system. Reflective signals associated with the gold nanoparticle core were visible, predominantly in the perinuclear region of treated macrophages, while the control (untreated) cells exhibited no such reflective patterns (Figure 5). These spots, discernible through reflective light microscopy, serve as optical signatures of the nanoparticulate presence and provide strong evidence of cellular uptake.

The internalization assessment was qualitative and based on a visual identification of the characteristic light-scattering signature of AuNPs. The presence of numerous and concentrated reflection spots within the cytoplasmic compartment—especially around the perinuclear region—implies a spatially organized internalization pattern. The observed localization, characterized by a dense accumulation near but not within the nucleus, supports the hypothesis that the nanostructures are being trafficked toward endosomal or proteasomal compartments, which are typically positioned in the perinuclear zone. Interestingly, the nuclear area itself remains devoid of such reflectance, reinforcing the idea of targeted subcellular localization.

The levels and intensity of fluorescent signal emission in the red spectrum were carefully evaluated following exposure of the macrophage cells to the nanostructured compound. The cultures treated with CEA-AuNP, 50 µg/mL, exhibited a markedly increased presence of distinct red fluorescent emission spots, as opposed to control groups exposed to unfunctionalized AuNPs (Figure 6).

This elevated fluorescent signal is indicative of enhanced proteasomal activity and indicates intracellular antigen processing. Although no quantitative fluorescence analysis was conducted in this study, the finding suggests that the nanostructured compound facilitates a more efficient delivery and subsequent intracellular routing of the antigen toward degradation and presentation pathways, particularly via the ubiquitin–proteasome system.

## 4. Discussion

The development of effective cancer vaccines hinges not only on the identification of suitable tumor-associated antigens but also on the optimization of delivery systems capable of enhancing antigen stability, presentation, and uptake by immune cells [5,10,18]. In this context, our findings provide insight into the structural and functional characteristics of a nanovaccine construct based on AuNPs functionalized with a CEA peptide [9]. The results presented in this study offer compelling evidence for the biocompatibility, internalization, and initial antigen processing potential of the CEA-AuNP system, proposed as a nanovaccine platform for tumor antigen delivery.

The results obtained from the synthesis and characterization of the nanostructures highlight the success of the Turkevich method in generating a stable nanostructured product with controlled dimensions and potential for further functionalization [19]. The presence of the absorption maximum of around 522 nm in the UV–Vis spectrum, together with the monodispersed distribution of around 20 nm, confirms the synthesis of small, homogeneous gold nanoparticles, as is characteristic of classical colloidal synthesis methods [20].

Spectroscopic analysis using UV–VIS revealed significant modifications in the absorbance profile following functionalization with CEA. While unmodified AuNPs exhibited the expected narrow surface plasmon resonance (SPR) peak centered at 522 nm, the functionalized construct demonstrated a broadened absorbance plateau ranging from 510 to 700 nm. This shift is consistent with changes in the local refractive index surrounding the nanoparticles, caused by the peptide corona [21]. Such spectral transformations are frequently associated with successful bioconjugation, as the protein or peptide layers alter the dielectric environment and introduce new absorbance contributions [22]. These findings support the hypothesis that CEA was successfully immobilized on the nanoparticle surface, creating a stable bio-nanoconjugate with modified optical properties [23].

Hydrodynamic diameter measurements further confirmed the efficiency of the functionalization process [24]. The observed increase in the average particle size from ~20 nm (unmodified AuNPs) to ~80 nm for the CEA-AuNP complex suggests the presence of a surface-bound organic layer, consistent with the addition of the antigenic peptide. The size range of ~80 nm is within the optimal window for uptake by antigen-presenting cells, such as macrophages and dendritic cells, which are instrumental in the initiation of adaptive immune responses [25,26]. Notably, despite the increase in particle size, the system retained a monodisperse distribution, indicating that aggregation was effectively prevented—a critical factor for biological stability, biodistribution, and immunological safety [27]. A mild broadening of the size distribution was noted, which is typical in protein or peptide surface conjugation, where variability in molecular loading across the nanoparticle population occurs [28,29].

While the current characterization employed UV–Vis spectroscopy, DLS, FT-IR, and AFM to confirm the size, morphology, and surface functionalization of the gold nanoparticles, we acknowledge the absence of high-resolution electron microscopy data (e.g., TEM or SEM). Such imaging would provide direct visual confirmation of nanoparticle shape and core size. This limitation will be addressed in future studies to further validate the nanoscale architecture of the CEA-AuNP construct.

The in vitro studies assessing CEA-AuNP cytotoxicity in the macrophage cell culture revealed high levels of cell viability and proliferation across all tested concentrations of the nanoconstruct, with values exceeding 85%—a benchmark widely recognized as acceptable for biomedical applications [30,31,32]. Functionally, this indicates that the nanoparticle system does not elicit cytotoxic responses in macrophages, even at upper dose limits. This aligns with prior studies reporting the minimal toxicity of peptide-coated AuNPs, reinforcing its suitability for biomedical applications [33,34]. From a nanovaccine perspective, high viability is essential to preserve antigen-presenting cell function during antigen delivery, ensuring effective immune engagement.

Although the results show a trend toward reduced proliferation at higher doses, the absence of formal statistical analysis limits conclusions about significance. Future studies will include full statistical evaluation. These data encourage further translational development, particularly for vaccine strategies requiring systemic or repeated administration, where long-term tolerance and minimal adverse effects are important [35].

Despite the known potential interactions of AuNPs with tetrazolium-based assays, the minimal optical interference was confirmed by including nanoparticle-only controls (without cells). Nevertheless, future studies will incorporate orthogonal viability assays, such as resazurin or LDH release, to validate MTT findings.

The analysis of apoptosis through caspase-3 activation further supports the notion of controlled cellular responses [36]. Caspase-3 is a key executioner protease in the apoptotic pathway, and its elevated expression or activation is a hallmark of programmed cell death [37]. We observe a direct proportionality between the nanomaterial concentration and the caspase-3 level. Functionally, this may reflect the early activation of programmed cell death pathways, potentially linked to nanoparticle uptake and intracellular stress responses. While excessive apoptosis would be detrimental, mild and localized apoptotic signaling—particularly in antigen-presenting cells—can enhance antigen cross-presentation and immunological priming, as previously observed in nanoparticle vaccine platforms [38,39].

The activation of caspase-3 observed in our study, with a predominantly perinuclear localization, is consistent with early events in intrinsic apoptosis. This pattern suggests that the nanoconstruct may initiate apoptosis through mitochondrial-mediated pathways, a mechanism critical for controlled cell death without inducing inflammatory responses [40,41,42]. Such targeted activation is advantageous in immunotherapeutic contexts, where mild apoptosis may assist in the maturation and antigen-presenting functions of dendritic cells and macrophages, ultimately enhancing T-cell priming [43].

While caspase-3 is a key effector in apoptosis, we recognize that a complete analysis would benefit from complementary methods such as Annexin V/PI staining or TUNEL assays. These will be integrated into future investigations to further validate the apoptotic profile.

Macrophage internalization studies revealed the uptake of the CEA-AuNPs, as evidenced by distinct reflection signals derived from the gold nanoparticle core. These reflectance-based signatures confirmed the intracellular presence and trafficking of the nanoconstruct, with a characteristic accumulation in the perinuclear area. This subcellular distribution aligns with the typical localization of proteasomal and endosomal compartments, which are responsible for the degradation and processing of antigenic peptides [44,45]. The nuclear region remained devoid of nanoparticulate signals, thus reinforcing the conclusion that the compound avoids genotoxic interaction and instead follows a defined intracellular processing route [46].

Fluorescence imaging, performed using rhodamine-filtered microscopy, revealed markedly elevated signal intensity in the macrophage cultures treated with the CEA-AuNPs compared to controls. This enhancement correlates with heightened proteasomal activity, which is essential for the processing of endogenous antigens and their subsequent loading onto major histocompatibility complex (MHC) class I molecules [47,48]. The red fluorescence signal, concentrated in the cytoplasm and especially enriched near the perinuclear zone, suggests the active engagement of the ubiquitin–proteasome pathway, a key element in effective antigen presentation and adaptive immune activation [49,50]. The intensity and widespread distribution of the red fluorescent signals suggest that peptide degradation and processing may be occurring in compartments associated with MHC class I loading, warranting further investigation. These findings collectively suggest that the CEA-AuNP system may play a dual role: as an efficient intracellular delivery vehicle and as a potent stimulator of antigen-processing pathways. While downstream immune responses such as cytokine release and T-cell activation were not assessed in this study, the observed antigen processing indicates early steps of intracellular immune engagement.

The intracellular and intracytoplasmic localization of the nanostructured compound, confirmed by both reflective and fluorescent imaging, supports its suitability for vaccine applications. This localization pattern ensures efficient internalization and routing for antigen presentation. The ability of the metallic gold core to serve as both a delivery scaffold and a detectable marker enhances the traceability of the compound during cellular interactions. This ensures that the antigenic payload is not only internalized but also appropriately directed for effective antigen presentation by macrophages, a critical step in the initiation of adaptive immune responses. This aspect is critical for future in vivo applications where biodistribution, retention, and immunogenicity must be quantitatively assessed.

In the broader context of cancer immunotherapy, these results highlight the potential of nanoparticle-based antigen delivery systems to overcome several limitations associated with conventional vaccine platforms, such as poor cellular uptake, antigen degradation, and suboptimal immune activation. By demonstrating safety, intracellular targeting, and efficient antigen processing in vitro, the CEA-AuNP formulation advances toward in vivo preclinical evaluation with a strong foundational profile.

Further investigations should include assessments of dendritic cell activation, antigen-specific T-cell responses, and tumor regression in animal models. Additionally, cytokine profiling, MHC surface expression analysis, and co-culture with tumor cells would provide a more complete picture of immunogenic potential. The conclusions of the data analyzed so far support the opportunity to continue the experiments in the macrophage- adenocarcinoma cell co-culture platform. Additionally, long-term biocompatibility and immunogenic memory will need to be addressed to fully elucidate the translational potential of this nanovaccine candidate.

Although fluorescence and dark-field imaging provided evidence of intracellular localization and antigen processing, these assessments were qualitative. The absence of quantitative image analysis (e.g., pixel intensity measurement or spectral mapping) is acknowledged as a limitation of the current study. Future investigations will incorporate automated tools such as ImageJ or CytoViva spectral mapping software to objectively quantify nanoparticle uptake and fluorescence intensity, ensuring robust spatial and comparative analysis. In addition, including larger sample sets and standardized imaging fields will help enhance the reproducibility and interpretability of visual data.

While the current study successfully demonstrates the intracellular delivery and antigen processing of the CEA-AuNP complex in macrophages, downstream immunological markers—such as pro-inflammatory cytokine release and MHC surface expression—were not included. These will be essential for determining whether antigen processing leads to effective immune activation. Future work will incorporate flow cytometric and ELISA-based analyses to assess these parameters in vitro and in vivo, offering a more complete immunological profile of the nanovaccine candidate.

## 5. Conclusions

The functionalized gold nanoparticles developed in this study demonstrate structural stability, favorable size for cellular uptake, and successful conjugation with the target peptide antigen. These physicochemical characteristics are of particular relevance to translational applications [51].

These findings represent an essential preclinical step in evaluating the therapeutic viability of nanostructured immunotherapies targeting cancer-associated antigens. In this case, the CEA-AuNP structure has the ability to function as an efficient platform for antigen delivery and processing in antigen-presenting cells while maintaining a favorable biocompatibility profile.

Future work will focus on assessing the immunostimulatory effects of this nanovaccine in relevant in vivo models and exploring its potential synergy with adjuvants or checkpoint inhibitors in the context of combinatorial cancer immunotherapy [52,53].

## Figures and Tables

**Figure 1 vaccines-13-00668-f001:**
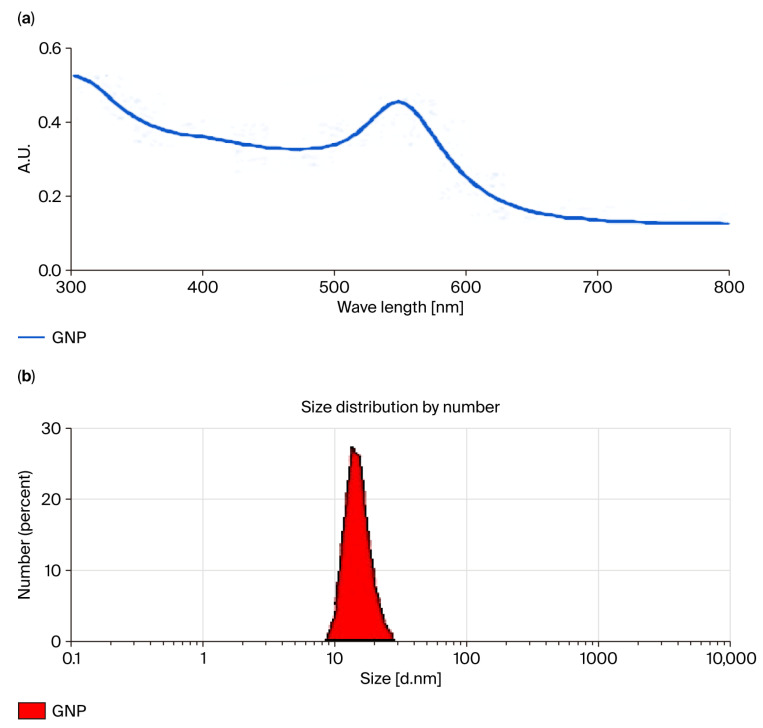
Characterization of unfunctionalized gold nanoparticles (AuNPs). (**a**) UV–Vis absorption spectrum showing a sharp surface plasmon resonance peak at 522 nm, indicating spherical morphology and high colloidal stability. (**b**) Dynamic light scattering (DLS) analysis confirming a monodisperse population with an average hydrodynamic diameter of ~20 nm and low polydispersity index (PDI).

**Figure 2 vaccines-13-00668-f002:**
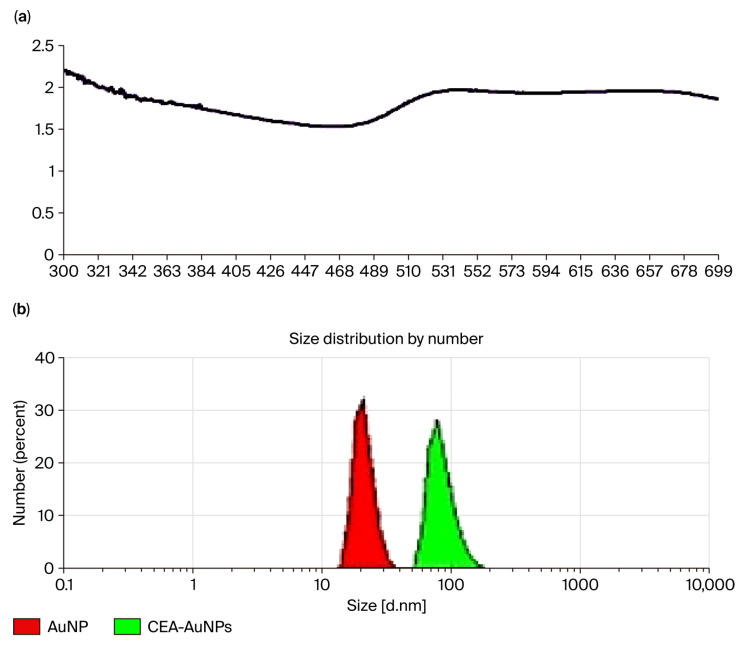
Characterization of CEA-functionalized gold nanoparticles (AuNP–CEA). (**a**) UV–Vis absorption spectrum of the functionalized nanostructure showing a broadened shoulder (510–700 nm), consistent with surface modification and biomolecule conjugation. (**b**) Dynamic light scattering (DLS) analysis showing particle size distribution by number for unmodified gold nanoparticles (AuNP, red) and CEA-functionalized gold nanoparticles (CEA-AuNPs, green). The red peak corresponds to a narrow monodisperse population centered around ~20 nm, characteristic of well-dispersed AuNPs. The green peak, representing CEA-AuNPs, shows a slight shift toward larger sizes (~80 nm), consistent with successful peptide conjugation increasing the hydrodynamic diameter.

**Figure 3 vaccines-13-00668-f003:**
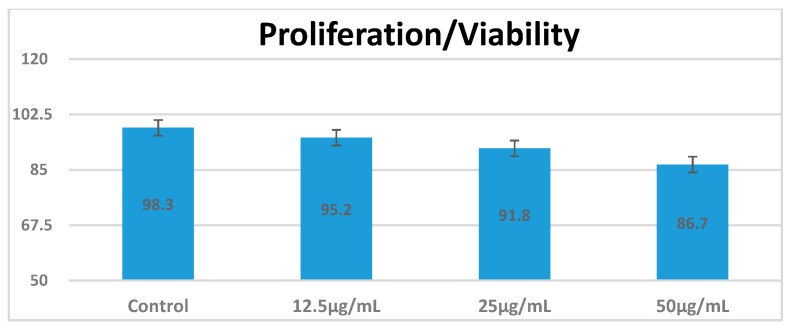
MTT assay results comparing cell viability of RAW 264.7 macrophages after 24 h exposure to various concentrations of CEA-AuNPs (12.5, 25, and 50 µg/mL). The control group represents untreated cells (PBS vehicle). Data are shown as mean ± SD from three replicates.

**Figure 4 vaccines-13-00668-f004:**
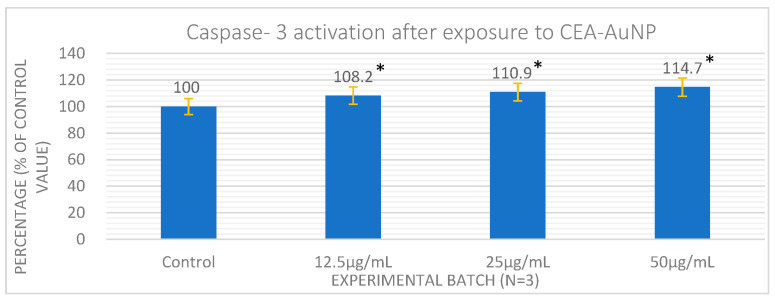
A direct proportionality is observed between the nanomaterial concentration and the level of caspase-3. Each value represents the mean of three independent experiments ± standard deviation. (* = *p* < 0.05).

**Figure 5 vaccines-13-00668-f005:**
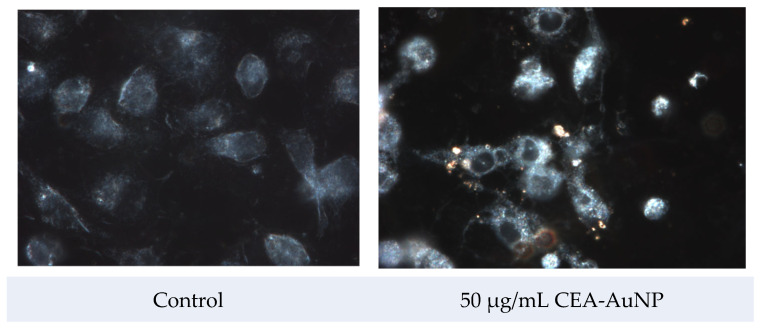
Dark-field hyperspectral microscopy images of macrophages exposed to CEA-AuNPs. The control group (PBS-treated) shows minimal light reflectance, while nanoparticle-treated cells display distinct reflectance signatures, indicating uptake. (Cytoviva, obj 400×).

**Figure 6 vaccines-13-00668-f006:**
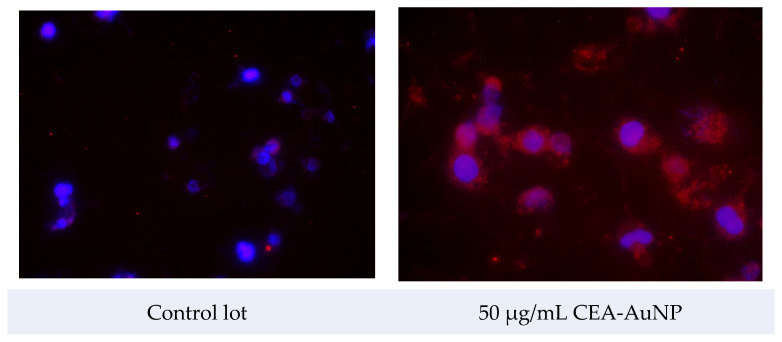
Fluorescence microscopy using Proteostat^®^ dye to detect intracellular protein aggregates associated with antigen processing. The PBS-treated group served as a control and shows minimal signal; nanoparticle-treated cells show enhanced fluorescent aggregation (FSX100 microscope with rhodamine filter, ob ×40).

## Data Availability

The raw data supporting the conclusions of this article will be made available by the authors on request.

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
