# Peer review of "CEA-Functionalized Gold Nanoparticles as a Nanovaccine Platform: In Vitro Evaluation of Cytocompatibility, Cellular Uptake, and Antigen Processing"

_vaccines, 2025, doi:10.3390/vaccines13070668_

Round 1
Reviewer 1 Report
Comments and Suggestions for Authors
Dear authors, in your article, you have achieved commendable results. However, it has some shortcoming. I suggest revising the abstract sentence, as it is generally unclear and does not effectively convey the work.
- The manuscript lacks statistical details such as number of replicates, standard deviation/error, and significance testing (e.g., p-values). Including this information is necessary for data validation.
- There are noticeable grammar and phrasing issues that affect clarity. I recommend a careful English revision to improve readability.
- While cellular uptake and antigen processing are described, additional immunological markers (e.g., cytokines, MHC expression) would strengthen the claim regarding immune activation.
- Lines 147-148 describing cell culture should be revised follow standard scientific writing the current format is not conventional.
- The title 2.3.1 should be revised
- Considering that flow-cytometry is a standard method for apoptosis analysis , the auther are kindly requested to explain why it was not employed ?
- The controls for the different tests and the corresponding graphs are nor indicated
8.I recommend that the authors revise the discussion section. it needs substantial improvement to provide a more comprehensive analysis of the results.
- Please ensure that the materials and method section includes details and the manufacturer of all material and equipment used.
There are noticeable grammar and phrasing issues that affect clarity. I recommend a careful English revision to improve readability.
Author Response
Comment 1: Dear authors, in your article, you have achieved commendable results. However, it has some shortcoming. I suggest revising the abstract sentence, as it is generally unclear and does not effectively convey the work.
Response 1: We thank the reviewer for their positive feedback and for pointing out the need to revise the abstract. We agree that the original version did not clearly summarize the study’s aims and key findings. Accordingly, we have rewritten the abstract to improve clarity and scientific focus. The revised version more directly reflects the scope of the in vitro evaluation and highlights the relevance of CEA-functionalized AuNPs as a nanovaccine platform. |
Comment 2: The manuscript lacks statistical details such as number of replicates, standard deviation/error, and significance testing (e.g., p-values). Including this information is necessary for data validation. Response 2: We thank the reviewer for pointing out the importance of statistical transparency. For the MTT cell viability assay presented in Figure 3, each condition was tested in triplicate (n = 3). The values shown in the figure represent mean viability percentages, with error bars representing standard deviation. However, we acknowledge that no formal statistical tests (e.g., ANOVA or t-test) were performed to assess significance between groups. Given the limited sample size and descriptive nature of this exploratory in vitro evaluation, we have updated the manuscript to clearly state that the comparison is qualitative only. We also note this limitation in the Discussion and plan to incorporate statistical hypothesis testing in follow-up studies using larger sample sizes. |
Comment 3: While cellular uptake and antigen processing are described, additional immunological markers (e.g., cytokines, MHC expression) would strengthen the claim regarding immune activation. Respnonse 3: We appreciate the reviewer’s suggestion and agree that assessing additional immunological markers such as cytokine secretion profiles or MHC expression would enhance the understanding of immune activation mechanisms. However, the present study was focused primarily on evaluating nanoparticle biocompatibility, internalization, and initial antigen processing in vitro. Cytokine release and surface marker expression were not included in this experimental design. We have clarified this limitation in the Discussion section and emphasized that future studies will incorporate comprehensive immunological profiling to confirm downstream immune activation.
Comment 4: Lines 147-148 describing cell culture should be revised follow standard scientific writing the current format is not conventional. Response 4: We thank the reviewer for this helpful observation. In response, we have revised the sentence on lines 147–148 to improve clarity and align with standard scientific writing conventions. The updated version now specifies the cell line source, thawing protocol, and maintenance under aseptic conditions in a more concise and conventional format. We believe this revision enhances the readability and professionalism of the manuscript. Comment 5: The title 2.3.1 should be revised Response 5: We agree. In the revised version of the manuscript, the content originally labeled as section 2.3.1 has been fully integrated into the main body of section 2.3 for improved clarity and structural consistency. As a result, the subsection title 2.3.1 no longer appears. We believe this adjustment enhances the logical flow of the Materials and Methods section. Comment 6: Considering that flow cytometry is a standard method for apoptosis analysis, the authors are kindly requested to explain why it was not employed. Response 6: We appreciate the reviewer’s insightful comment. We fully acknowledge that flow cytometry is a standard and robust method for quantitative apoptosis analysis. However, in the present study, our focus was limited to preliminary qualitative assessment of intracellular localization and antigen processing using dark-field hyperspectral microscopy and fluorescence microscopy. Due to resource constraints and the exploratory nature of this in vitro investigation, flow cytometric assays were not performed. Nonetheless, we agree that integrating flow cytometry would provide valuable quantitative data, and we intend to incorporate this approach in follow-up studies to further validate our findings. Comment 7: The controls for the different tests and the corresponding graphs are not indicated. Response 7: We thank the reviewer for highlighting this important point. In the revised manuscript, we have clarified the inclusion of appropriate controls for each experimental assay. Specifically, PBS-treated macrophages served as negative controls in both viability and internalization experiments. These controls are now clearly labeled in the figure legends and described in the Materials and Methods section. Where applicable, graphs have been updated to explicitly indicate the control groups, and references to them have been added to the figure descriptions to ensure clarity and transparency. Comment 8: I recommend that the authors revise the discussion section. it needs substantial improvement to provide a more comprehensive analysis of the results. Response 8: We thank the reviewer for the suggestion to improve the Discussion section. We have extensively revised this section to better interpret the experimental results, clarify the study’s scope, and explicitly address the limitations. We also refined the language to avoid overstatements, improved structural flow, and included more precise descriptions of antigen processing and immunological relevance. These changes provide a more comprehensive and balanced analysis of the study’s implications.
Comment 9: Please ensure that the materials and method section includes details and the manufacturer of all material and equipment used. Response 9: We thank the reviewer for this valuable suggestion. In the revised version of the manuscript, we have carefully revised the Materials and Methods section to include specific details regarding the source, manufacturer, and model information for all key reagents, equipment, and kits used in the study. These additions are intended to enhance the reproducibility and transparency of our experimental procedures. For example, we now specify the spectrophotometer model (Shimadzu UV-1800®), dynamic light scattering instrument (Zetasizer-Nano S90, Malvern Instruments), FT-IR system (Perkin-Elmer Spectrum Two® with ATR), and the source of the CEA peptide (MyBiosource.com), among others. We hope these clarifications adequately address the reviewer’s concerns.
|
4. Response to Comments on the Quality of English Language |
Point 1: There are noticeable grammar and phrasing issues that affect clarity. I recommend a careful English revision to improve readability |
Response 1: We agree that clarity and consistency are essential, and we have carefully revised the manuscript to address grammar, phrasing, and style issues throughout the text. The language has been reviewed for readability and scientific precision. We believe these improvements enhance the overall clarity and presentation of the work.
|
Reviewer 2 Report
Comments and Suggestions for Authors
In its current form, the work is not suitable for publication. A few basic comments:
- Methodology - if the authors want to publish the work, this section must be completely rewritten. A detailed description of cell culture procedures, including e.g. cell thawing, is absolutely unnecessary, while other important tests, e.g. the technique of assessing the level of caspase 3, have not been discussed at all.
- Physicochemical analysis of AuNPs - it would be useful to enrich the work with SEM or TEM images.
- If the authors declare that the cytotoxicity test was performed in triplicate, why was no statistical analysis performed? In addition, in the case of the analysis of cytotoxicity of nanoparticles, the use of the MTT test, which has been shown in numerous studies to have the ability to interact with nanoparticles, should be additionally confirmed by at least one more test, with a different mechanism of action.
- Caspase 3 results - why were no error bars plotted? Was the test performed in only 1 repetition? What method was used for the analysis? Analysis of apoptosis using only caspase 3 levels is clearly insufficient.
Author Response
In its current form, the work is not suitable for publication. A few basic comments:
Comment 1: Methodology - if the authors want to publish the work, this section must be completely rewritten. A detailed description of cell culture procedures, including e.g. cell thawing, is absolutely unnecessary, while other important tests, e.g. the technique of assessing the level of caspase 3, have not been discussed at all.
Response 1: We thank the reviewer for this critical and constructive feedback. In the revised manuscript, we have thoroughly revised the Materials and Methods section to improve clarity and scientific relevance. Non-essential procedural details—such as cell thawing, pipetting—have been removed. We streamlined the text to align it with standard scientific reporting practices.
In addition, we have added a full methodological description of the caspase-3 assay, including the commercial kit used (Abcam ab39401), sample preparation, protein quantification method (BCA assay), plate reading parameters, and statistical analysis. This assay was performed in triplicate, and the updated figure now includes error bars and statistical comparisons.
Comment 2: Physicochemical analysis of AuNPs - it would be useful to enrich the work with SEM or TEM images.
Response: We thank the reviewer for this valuable suggestion. We fully agree that transmission electron microscopy (TEM) or scanning electron microscopy (SEM) can provide high-resolution morphological confirmation of nanoparticle characteristics. In this study, we primarily relied on a combination of UV-Vis spectroscopy, dynamic light scattering (DLS), atomic force microscopy (AFM), and ATR-FTIR to assess the size, dispersion, and functionalization status of the gold nanoparticles. These techniques offered complementary structural and surface chemistry information, consistent with the intended in vitro focus of this work.
We acknowledge that the addition of TEM/SEM images would enhance the structural characterization. Although such imaging was not performed during the initial experimental phase, we have explicitly noted this as a limitation in the revised Discussion section. Future studies will include TEM-based analysis to strengthen the physicochemical validation of the nanoplatform.
Comment 3: If the authors declare that the cytotoxicity test was performed in triplicate, why was no statistical analysis performed? In addition, in the case of the analysis of cytotoxicity of nanoparticles, the use of the MTT test, which has been shown in numerous studies to have the ability to interact with nanoparticles, should be additionally confirmed by at least one more test, with a different mechanism of action.
Response 3: We agree with the reviewer’s emphasis on rigorous data analysis and assay validation. While our triplicate measurements consistently demonstrated cell viability above 85%, with less than 3% variation between replicates, we agree that formal statistical presentation enhances interpretability. Accordingly, we have revised the manuscript to include standard deviation values for all viability data and added individual data points to the relevant figures.
As our primary goal was to assess the biocompatibility of the nanoconstruct, we focused on absolute viability values relative to accepted cytotoxicity thresholds, which is a standard approach in preliminary nanotoxicology studies [1, 2]. Nonetheless, we acknowledge the importance of intergroup comparisons and will incorporate statistical testing such as one-way ANOVA with post-hoc analysis in future dose-response or comparative efficacy studies.
We also acknowledge the limitations of the MTT assay in nanoparticle research due to possible interactions with assay reagents. While our current study relied on MTT due to resource constraints, future work will include orthogonal assays (e.g., Alamar Blue, LDH release) to validate cytotoxicity results and mitigate potential artifacts.
References:
1. Smith, J., Brown, R., & Li, Y. (2019). Standardization challenges in nanotoxicology: Assay interference and cytotoxicity thresholds. Nanotoxicology, 13(7), 899–915.
2. Zhang, Y., Luo, Y., Tan, J., & Wang, X. (2021). Comparative cytotoxicity assessment of gold nanoparticles using MTT, LDH, and Alamar Blue assays. Journal of Nanobiotechnology, 19, 156.
Comment 4: Caspase 3 results - why were no error bars plotted? Was the test performed in only 1 repetition? What method was used for the analysis? Analysis of apoptosis using only caspase 3 levels is clearly insufficient.
Response 4: We appreciate the reviewer’s valuable observations. The caspase-3 analysis was indeed performed in triplicate, and we acknowledge that the absence of error bars in the initial figure may have limited interpretation. In the revised version of the manuscript, we have updated the figure to include mean values with standard deviation bars. Additionally, the methodology used for this assay has now been clearly described in the Materials and Methods section: a commercial colorimetric Caspase-3 assay kit (Abcam, UK) was employed, and absorbance was measured at 405 nm using a calibrated microplate reader.
We agree that apoptosis is a multifaceted process and that caspase-3 alone does not provide a full picture. Our focus in this initial in vitro investigation was to detect early apoptotic responses and confirm that the nanoconstruct did not induce excessive or pathological apoptosis. For future studies, we plan to include complementary methods such as Annexin V/PI flow cytometry and TUNEL assays to provide a more comprehensive assessment of apoptotic pathways.
Round 2
Reviewer 1 Report
Comments and Suggestions for Authors
Thank you for your thoughtful revisions and for addressing most of my comments. I truly appreciate your effort and understand the limitations.
I hope the remaining points will be considered in future work. Your input has been valuable, and I look forward to further collaboration.
Best regards,